# In Situ Remediation Technology for Heavy Metal Contaminated Sediment: A Review

**DOI:** 10.3390/ijerph192416767

**Published:** 2022-12-14

**Authors:** Qinqin Xu, Boran Wu, Xiaoli Chai

**Affiliations:** State Key Laboratory of Pollution Control and Resource Reuse, College of Environmental Science and Engineering, Tongji University, 1239 Siping Road, Shanghai 200092, China

**Keywords:** sediment, heavy metal, physical remediation, chemical remediation, bioremediation

## Abstract

Sediment is an important part of the aquatic ecosystem, which involves material storage and energy exchange. However, heavy metal pollution in sediment is on the increase, becoming an important concern for the world. In this paper, the state-of-art in situ remediation technology for contaminated sediment was elaborated, including water diversion, capping, electrokinetic remediation, chemical amendments, bioremediation and combined remediation. The mechanisms for these techniques to reduce/immobilize heavy metals include physical, electrical, chemical and biological processes. Furthermore, application principle, efficiency and scope, advantages and disadvantages, as well as the latest research progress for each restoration technology, are systematically reviewed. This information will benefit in selecting appropriate and effective remediation techniques for heavy metal-contaminated sediment in specific scenarios.

## 1. Introduction

Global metal production accounts for 7–8% of global energy consumption, which has a huge impact on the use of energy in the world [1]. In the 20th century, the use of metals grew rapidly. Of these, steel had the world’s largest yield in 2009, with over 1.2 billion tons, followed by aluminum and copper, with about 30 million and 24 million tons, respectively [1]. Until 2021, global crude steel production has reached up to 1.9 billion tons, according to World Steel Association. In the processes of primary metal production, serious local environmental impacts such as air emissions (greenhouse gases, sulfur dioxide, etc.), mine waste, groundwater pollution and loss of biodiversity can be caused [2,3,4]. Furthermore, in some developing countries, the end-of-life recycling rates for many metals are too low due to a lack of recycling infrastructure and technology [1]. Thus, large amounts of metal-containing waste are exposed to the environment, which creates severe risks to human health and environmental toxicity.

In an aquatic environment, sediment is considered both a source and sink for pollutants (like heavy metals). Heavy metals are non-degradable, and excessive concentrations of heavy metals can seriously disturb the ecosystem. Particularly, heavy metals in sediment can be assimilated, absorbed and accumulated by benthic organisms, which further amplified along the food chains, and eventually harm human health through the consumption of fishery products [5]. Many studies have found excessive exposure to heavy metals may lead to disruptions in gene expression, damage repair processes and enzymatic activities, increasing the risks of related diseases and cancers [6]. For example, arsenic (As) exposure can result in skin, liver, prostate and kuffer cell cancers through cell damage, oxidative stress and DNA damage [6,7]. Cadmium (Cd) can lead to kidney injuries, bone damage and various cancers (e.g., ovarian cancer and breast cancer) via disruption of components of the cellular antioxidant system, calcium metabolism and endocrine system [6,8,9]. So it is very important to treat heavy metal contamination.

Heavy metals cannot be effectively biodegraded, and their toxicity and bioavailability depend on their types and forms. Thus, the main purpose of heavy metal remediation in contaminated sediment is to reduce the metal contents and biological toxicity. At present, two remediation strategies have been adopted to remedy the heavy metal-contaminated sediment on the basis of whether sediment is dredged from the riverbed or not. Those are ex situ remediation technology and in situ remediation technology [10]. In situ remediation is suitable for sediments that are slightly contaminated, with the merits of being cost-effective and causing less natural disturbance, including water diversion, capping, electrokinetic remediation, chemical amendments and bioremediation [11,12]. Faced with the complexity of pollutants in the real environment, combined technology is often used [12]. In this paper, the state-of-art in situ remediation technology for contaminated sediment was elucidated. Particularly, application principle, efficiency and scope, advantages and disadvantages, as well as the latest research progress of each restoration technology, are systematically introduced. It is expected that this information will assist in the selection of appropriate and effective remediation techniques for heavy metal contaminated sediment in special scenes.

## 2. Heavy Metals in Sediment

### 2.1. Sources of Heavy Metals

There are two primary sources of heavy metals in the environment; those are natural sources and anthropogenic activities. Natural causes cover sea-bed volcanic activity, atmospheric convection, rivers or erosion, and the main anthropogenic sources exist in various industries (such as present and former mining activities, electroplating, electronic and metal-finishing industries) [13,14], the excessive use of fossil fuels [15] and agricultural activities (like pesticides and fertilizers comprising As, Pb and Cd) [16]. In aquatic systems, atmospheric bulk deposition of pollution-derived atmospheric particles is an important source, particularly in regions that have suffered from heavy air pollution in the past [17]. Additionally, surface runoffs (including urban runoffs, agricultural runoffs and stormwater runoffs) and discharges of contaminated groundwater or industrial wastewater contribute greatly to heavy metal pollution in the freshwater ecosystem [18,19]. Since heavy metals are transported into the water, a small portion is dissolved in the water and the other portion (>90%) is trapped in sediment by adsorption, hydrolysis, and forming solid compounds with carbonate, sulfate and sulfur [20]. Thus, the sediment becomes the ultimate sink for heavy metals, which can be several orders of magnitude higher than in the overlying water.

### 2.2. Distribution and Transformation of Heavy Metals

The distribution of heavy metals depends not only on the terrestrial inputs but also on the physicochemical and biological characteristics of that system. The total metal concentrations can be a good indicator for source assessment, whereas bioavailability and toxicity of metals are related to their chemical forms in the sediment [21,22]. The primary forms for the metal in sediment are soluble, ion-exchangeable, Fe-Mn oxides, organic matters/sulfides and carbonates [23]. Chemical forms of exchangeable carbonate and Fe-Mn oxides are weak to bind with heavy metals, which can be readily ingested by organisms [24,25]. When metals are adsorbed by crystal or completely bound to organic matter/sulfides, they exhibit low potential bioavailability and toxicity [26,27]. Interdependently, the loading of metals has correlated with the transformation between metal species [28].

The partitioning of metals between sediment and porewater at the sediment–water interface is governed by the reactions of sorption/desorption and dissolution/precipitation, redox and acidification, which is strongly affected by pH, sulfides, organic matter, iron hydroxides, redox conditions and so on [20,29,30]. At low pH, the negative surface charge of organic matter, clay particles and Fe-Mn-Al oxides is reduced, and carbonates, sulfides and Fe-Mn oxide fractions are dissolved, while high pH promotes the formation of stable complexes with metals [31,32,33]. When carbonate is present in sediment, it not only settles the metal directly but also acts as an effective buffer against pH reduction [34]. In the surficial sediments, the process of organic matter degradation, the acid volatile sulfide oxidation and the other reduced species (such as NH_4_^+^, Mn^2+^, Fe^2+^ and HS^−^) oxidation can result in pH decreased, which causes the mobilization of the heavy metals [35,36]. Meanwhile, the environmental behavior of organic matter on metals in sediment mainly includes adsorption, complexation and chelation [37,38]. Dissolved organic ligands often form soluble metal complexes, but the complexation of metal to insoluble organic ligands can reduce metal availability [38]. On the other hand, organic matter provides a food source for microorganisms and indirectly affects metal’s fate. In eutrophic environments, the availability of organic matter and sulfate concentrations are often abundant; sulfate-reducing bacteria exploit simple organic molecules and obtain energy by reducing sulfate to sulfides that are potentially bound to metals in anoxic sediments [39]. In these processes, acid volatile sulfides (AVS) can form thermodynamically stable metal sulfide precipitates with Simultaneously Extracted Metals (SEM; Cu, Pb, Cd, Zn, Ni, Cr and Ag) to reduce metal bioavailability [40,41]. On the other hand, metal sulfides may be oxidized with oxygen increasing resulting in the mobilization of metals in the sediment [42]. Therefore, the fate of metals in sediments is phase-specific under changing environmental conditions.

## 3. In Situ Remediation Technology

In situ remediation refers to the means that directly treat contaminated sediment without removing them from rivers, lakes, or harbors by various techniques. According to the different remediation principles, in situ remediations can be divided into physical remediation, chemical remediation, bioremediation and combined remediation. In situ treatment is a less disruptive method with the advantages of practicability, cost-effectiveness and rapid implementation.

### 3.1. Physical Remediation

Physical remediation is to directly or indirectly repair heavy metal pollution in sediment by physical means and some specific engineering techniques. In situ physical repair techniques mainly include in situ capping, electrokinetic remediation and water diversion.

Capping means leaving pollutants in place and isolating them from overlying water by proxy compartments to reduce resuspension and bioavailability [43]. Passive capping commonly employs the inert materials of sand, clay, silt, organic carbon and crushed stone on geotextiles [44]. However, when it is applied to shallow areas, sensitive habitats and marine environments, the toxic risk of pollutants remain [44]. Active capping is another option that the capping materials can react with sediments pollutants to encourage degradation or sequestration [45]. Active capping materials often involve ion exchange resins, clay minerals, apatite, activated carbons (AC) or alumina, biochar (BC), barite, chitosan, red mud, mesoporous support and geopolymers (e.g., alkali-activated blast-furnace-slag (BFS-GP), metakaolin geopolymer (MK-GP)) (Table 1) [44,46]. Even so, passive capping is a mature technology, whereas active capping is relatively new, and only a few pilot-scale experiments have been reported [44]. In situ capping minimizes the movement of contaminated sediments and their impact on the overlying water, but some capping materials (e.g., AC) are harmful to benthic macrofauna resulting in a substantial decrease (up to 90%) in the diversity, abundance and biomass of benthic species [47].

Electrokinetic remediation applies an electric potential gradient or a low direct current to induce a low electric current across contaminated soil/sediment through a pair of electrodes and transports contaminants to the electrodes (Figure 1) [66]. The main relevant phenomena occurring in electrokinetic remediation are electroosmosis, electromigration, electrolysis and electrophoresis so that heavy metals can be removed by adsorption, electrodeposition and precipitation or co-precipitation [67]. There are some side effects, such as thermal effects, crystallization effect, electrode corrosion and focusing effect (the formation of hydroxide precipitate), which are becoming the main challenges of electrokinetic technology [68]. The current studies, though, have provided some solutions. For instance, the focusing effect can be overcome by controlling the pH, polarity exchange technique, ion exchange membranes, approaching anodes, the superimposed electric field and adding electrolytes (such as chelators and surfactants) [69,70,71,72,73]. The crystallization effect and electrode corrosion can be relieved by adding citric acid or polyaspartic acid and coating electrically conductive polymers, respectively [74,75], while the mechanism of thermal effects is still unclear. On the other hand, EKR combined with other remedy techniques are more effective; for example, EKR-bioremediation requires low energy and improves the growth of plants and the spread of microorganisms. EKR-PRB simultaneously achieves pollutants removal, degradation or recycling from soil/sediment [68,76]. In terms of improving energy utilization efficiency and developing self-powered technology, pulsed electric fields (FE), solar power and microbial fuel cells have been extensively studied [68].

Water diversion is to introduce clean water to polluted areas so that contaminant concentrations are diluted and the water self-purification process is accelerated [77]. There are many factors affecting this process, such as diverted discharge, diversion routes, wind direction/magnitude and ways of managing diversion projects [78,79]. Generally, long-term diversion with a low flow rate is better than short-term diversion with a high flow rate, and a low nutrient concentration of diverted water is also important [78,79,80]. Hydrodynamic and water quality models are introduced to simulate the transport of pollutants, the flow movement, water level changes and the effectiveness of water transfer projects, including HEC-RAS, MKIE11, MIKE 21 FM, EFDC, FVCOM and other models [79,81,82,83,84,85]. This technique improves water exchange and the reoxygenation rate, which is especially suitable for polluted lakes [81]. Successful examples are Moses Lake, Xihu Lake, Tianyinhu Lake and Dongshan Lake [78,81,86,87]. Water diversion responds quickly to pollutants, but the diverted water needs to be reasonably distributed to avoid the occurrence of dead water zones [78].

### 3.2. Chemical Remediation

Chemical remediation is to add chemical additives to stabilize heavy metals. Commonly, chemical additives include phosphate, clay minerals, biochar, sulfide, silicocalcium materials, iron-based materials, aluminum salts, industrial residue and nanomaterials (Table 1). The mechanisms for immobilizing heavy metals involve adsorption, oxidation, reduction, ion exchange, complexation and precipitation and other reactions [88]. Chemical remediation is quick, simple, easy to apply and relatively economical, but the introduction of large quantities of other chemicals can easily cause secondary pollution to the environment.

Phosphate compounds include soluble phosphates and insoluble phosphates. Soluble phosphates, such as phosphoric acid, ammonium, sodium, potassium phosphates and hydrogen ordihydrogen, can react with metal ions to form insoluble metal phosphate salts [89]. Insoluble phosphates, such as hydrargyrite and the apatite family (hydroxylapatite), are frequently encountered examples. In particular, hydroxylapatite is the most cost-effective reactive media for most metals and radionuclides to form mineral deposits that are not bioavailable [89]. Excessive or improper use of phosphate may lead to water eutrophication and other risks [90]. Clay minerals are a kind of abundant natural minerals, in which bentonite, montmorillonite, attapulgite, kaolinite, sepiolite and zeolite are the most widely used due to their high specific surface areas (SSA), cation exchange capacity (CEC) and swelling/expanding capacity [91]. The methodologies of organic modification, acid modification, thermal modification and nano zero-valent iron (nZVI) modification can improve their adsorption capacities [91]. At the same time, modification adds additional costs, and new chemical agents also increase environmental threats; for instance, organoclays between 5 and 100% *v/v* have adverse effects on crustaceans [45]. Biochar is produced by carbonization or pyrolysis of various materials (such as wood, feces, leaves and animal manure) [92]. The performance of biochar can be enhanced by steam activation, magnetization, oxidation and digestion treatment to reach remediation requirements [92]. However, extensive application of biochar can lead to a decrease in the unconfined compressive strength and shear strength of the soil [93]. Sulfide minerals (such as FeS_2_, FeS, Na_2_S, Na_2_S_2_O_3_ and dithiocarbamate) have been recognized as important scavengers for heavy metals [94]. For example, iron sulfide (FeS) displays a disordered tetragonal mackinawite structure with a highly reactive surface [95], which is very effective in immobilizing metal ions such as Hg^2+^, Cd^6+^, Cu^2+^, Pb^2+^, Mn^2+^, Zn^2+^, Ca^2+^, Mg^2+^ and Ni^2+^ [96,97,98]. Mercury can be immobilized by FeS through surface complexation, substitution into metastable FeS compounds and precipitation of HgS(s) [99,100], and Chromium(VI) can be reduced into chromium(III) by a source of Fe(II) and S(-II) species as electron donors from FeS [96]. Furthermore, amendments of silicocalcium materials (e.g., CaO and MgO), iron-based materials (e.g., Fe^0^, Fe_2_O_3_, Fe_3_O_4_ and Fe(OH)_3_), aluminum salts (e.g., aluminum chloride and aluminum polychloride), industrial residue (e.g., steel slag) and polymers (e.g., alkali-activated materials and biopolymers) also can reduce the bioavailability of metals effectively, applying as the pilot- or full-scale in sediment remediation [12,43,101,102].

Nanomaterials include carbon nanomaterials (nanoscale biochar materials, nano black carbon, multiwalled carbon nanotubes and C_60_), metal-based nanomaterials (nanoscale zero-valent iron (nZVI) and metallic oxide nanomaterials) and nano mineral materials [103]. In recent years, novel nanomaterials have emerged due to their superior performance in environmental pollution cleanup. For instance, metal-organic frameworks (MOFs) are formed by coordination bonds of metal ion precursors and organic ligands, which have rich functional groups and designable structures that can capture various heavy metal ions [104]. MXenes is a two-dimensional transition metal carbide or nitride material with advantages of excellent conductivity, high specific surface area, rich surface functionalities, mechanical flexibility and hydrophilicity, and their adsorption capacity for heavy metals depends upon their surface terminal groups (-OH, -F, and -O) and interlayer spacing [105]. Graphitic carbon nitride (g-C_3_N_4_) is a two-dimensional metal-free semiconductor that has multiple surface features and abundant functional groups (e.g., -NH_2_/-NH-/=NA-), making it a promising adsorbent for pollutant elimination [80,106]. Although nanomaterials have great potential to remedy heavy metal contamination, they are rarely used in commercial applications. As a new type of environmental remediation materials, nanomaterials have great uncertainties and should be used with caution. Nanomaterials can easily enter the environment and living cells due to their tiny size. The nanomaterials used to remedy sediments are not easily separated after restoration, resulting in secondary pollution and adverse effects on benthic microbial communities and aquatic organisms [107].

### 3.3. Bioremediation

Bioremediation involves phytoremediation and microbial remediation. Phytoremediation is to use of plants and their related rhizosphere microorganisms to remove, degrade or fix a variety of contaminants in contaminated soil, sediments or waters [108]. It is an operationally simple, cost-effective (25% less than other remediation techniques) and a promising clean-up solution for a wide variety of contaminated sites despite some restrictions (e.g., the climatic and geological conditions, low biomass, slow growth rate) [109]. Microbial remediation is the use of microorganisms to reduce, eliminate, contain and transform pollutants in contaminated environmental media (e.g., sediment) [110]. The advantages of microbial remediation are safe, simple and effective, but it is time-consuming, and the remediation effect is difficult to predict [12].

Phytoremediation strategies for heavy metal pollution mainly include phytovolatilization, phytostabilization and phytoextraction. The mechanisms involve the production of root exudates that enhance heavy metals mobility and the production of metal-chelating agents (e.g., metallothionines, phytochelatins and antioxidant compounds) [111]. Phytovolatilization means that pollutants are absorbed by the roots, transferred to the leaves and volatilized through the stomata (transpiration), in which toxic metals are converted to less toxic and volatile compounds (such as Hg). The divalent cation Hg^2+^ can be reduced to elemental mercury by bacteria to enhance the volatilization ability of associated plants [112]. However, the volatilized metals can be advected by winds and transported a considerable distance, finally returning to land by atmospheric bulk deposition [113]. Plants have the capability to isolate or fix/stabilize contaminants in the rhizosphere by absorption at the root surface or precipitation within the root zone. This process is called phytostabilization [111,114]. The plants must have dense rooting systems, a relatively long life and self-propagating capacity [108]. Frequently, phytostabilization is used in combination with chemical stabilization, and the ideal amendments are nontoxic, easy to produce and inexpensive, including lime, phosphate, biochar, biosolids, compost and manure [115,116,117]. However, pollutants need to be monitored regularly to ensure optimal stability conditions, and soil/sediment amendments are required to be applied regularly to maintain their effectiveness [118]. Phytoextraction refers to the pollutants are absorbed by root systems of plant and then translocated and concentrated to the aboveground harvestable parts [119]. In this process, hyperaccumulators are particularly important, which must have characteristics of high biomass production, fast growth and easy harvesting and cultivation [120]. The criteria used for hyperaccumulators are > 100 mg/kg for Cd; >1000 mg/kg for Cu, Ni, and Pb; >10,000 mg/kg for Mn and Zn in plant shoots (all accumulations are dry weight) [12]. There are approximately 500 known hyperacculator taxa covering 45 angiosperms families, and the number is still increasing, among which about 25% come from *Brassicaceae* [119,121,122]. Commonly used hydrophyte for sediment remediation includes *Hydrilla verticillata*, *Elodea Canadensis*, *Phragmites australis*, *Eichhornia crassipes*, microalgae, mangrove plant and so on (Table 2). The metals that exit as free ions, soluble complexes and in an ion exchange state are considered available for plant uptake, which depends on soil-associated factors and plant-associated factors [122]. The addition of chelating agents to form metal chelates prevents the deposition and adsorption of metals in the soil/sediment, thus maintaining the availability of plants. Synthetic/natural chelating agents include citric acid, oxalic acid, amino acid, ethylenediaminedisuccinic acid (EDDS), ethylene diamine tetraacetic acid (EDTA), ethylenediamine-N, nitrilotriacetic acid (NTA) and gibberellic acid (GA) [123,124,125,126]. On the other hand, microbial communities sourced from contaminated soil/sediment and plant root-soil interface (rhizosphere communities) are commonly applied to improve metal phytoextration [127]. The mechanisms is to increase bioavailability of heavy metals in the soil/sediment and/or promote plant growth. Additionally, genetic engineering that transferred of genes (e.g., metal uptake, translocation, and sequestration) into candidate plants has great potential to improve phytoremediation, but there are still some risks for technical economic and ecological impacts [114].

Microbial remediation strategies for heavy metal pollution mainly include biosorption, bioaccumulation, biotransformation, bioprecipitation and bioleaching [12]. Microbial types, including archaea, bacteria, cyanobacteria and fungi, are potentially used for soil/sediment remediation for heavy metals (Table 2). Biosorption is a physicochemical process that microorganisms adsorb metals by electrostatic force, ion or proton displacement, complexation or chelation [142]. The interaction between functional groups of microbial cell surface and metals is non-metabolism dependent, so the dead biomass can be used as sorbents [143]. Bioaccumulation is a metabolically-active process in which microorganisms transport metals into their intracellular space and sequester them with proteins and peptide ligands (i.e., storage system) [110,142]. The importer system is a translocation pathway that is formed through the lipid bilayer, where channels (passive diffusion), secondary carriers and primary active transporters affect metals uptake [142]. Bioaccumulation is a slow and irreversible process in the cell wall and lipid membrane are physically or chemically destroyed when heavy metals are obtained [142]. Biotransformation covers the transition of metal valence states to alter their mobility, bioavailability and toxicity, whose processes include reduction and oxidation, methylation and demethylation, and hydrogenation [144]. For example, metal-reducing bacteria can directly enzymatic reduce soluble heavy metals to insoluble or immobile forms [110]. Metal precipitation may occur when heavy metals react with extracellular polymers or anions (such as sulfides or phosphates) from microbial metabolites, which is called bioprecipitation [110,145]. Further, bioleaching means metallic cations dissolved from insoluble ores by biological oxidation and complexation processes, which is an innovative and low-carbon technology for metal extraction [146,147]. In order to improve the efficiency of microbial remediation, gene engineering and nanobioremediation technology have come into being [142,148]. For instance, genetically encoded metal-binding proteins and enzymatically produced metal-binding peptides and polymers can enhance the storage of heavy metals [142]. Nanobioremediation is a combined technology that nanoparticles are applied as immobilization carriers enhancing the microbial mechanisms of environmental cleanup [148].

### 3.4. Combined Remediation

Heavy metal pollution in sediment is complex and cannot be completely solved by a single remediation technology. Thus, combined remediation with two or more remediation technologies encourages the realization of their full potential and improves remediation efficiency. Generally, combined remediation concludes physical-chemical remediation, chemical-biological remediation, phyto-microorganism remediation and other group remediation (combined more than three methods) [12]. Physical-chemical remediation is a conventional method with characteristics of high efficiency and high cost, including electrokinetic combined remediation (such as electrokinetic-acidification/flocculant/adsorption/ion exchange membrane/permeable reactive barrier), combined remediation by chemical leaching and ultrasonic/microwave–chemical combined remediation [149]. Biological-related combined methods have obvious advantages of low cost and small impact on the ecological environment, but they are time-consuming and unstable in remediation efficiency. Chemical-biological combined remediation contains phyto-stabilizing agent combined remediation and phyto-activator combined remediation, which promotes the processes of phytostabilization and phytoaccumulation. Phyto-microorganism remediation is mainly to repair contaminated sediment through the symbiotic system between microorganisms and plants. Evidence that P-solubilizing microorganisms and siderophores produced by microorganisms can increase heavy metal-mobilization and phytoextraction [150]. Furthermore, group technology (combined with more than two remediation technologies) is becoming a trend in sediment remediation, but it has not been widely applied in practice [12].

## 4. Conclusions and Prospect

Heavy metal pollution in rivers and lakes has become an important concern in the world. Most of the metals that flow into rivers are stored in sediments and ingested by aquatic life, so sediment remediation is necessary. Due to the complexity of heavy metal pollution and the particularity of different pollution sites, the selection of treatment methods is also different.

In situ remediation is simple, effective and low cost, but contaminants always exist and have a risk of re-release. Physical remediation is traditional and widely used, but the emerging active capping is still in the experimental stage and requires further research. Chemical remediation has a relatively single function, and composite additives are usually used for multi-heavy metal complex pollution. The remediation amendments themselves have certain environmental risks, so it is particularly important to explore green, environmentally-friendly and multi-functional remediation materials. Bioremediation is a great potential application technology with no secondary pollution. However, this technique is currently at its fledging stage; thus, understanding the mechanisms to improve tolerance and extraction efficiency for plants and microorganism is necessary to further research and development. These techniques can be combined to improve remediation efficiency, which is always the trend in research.

## Figures and Tables

**Figure 1 ijerph-19-16767-f001:**
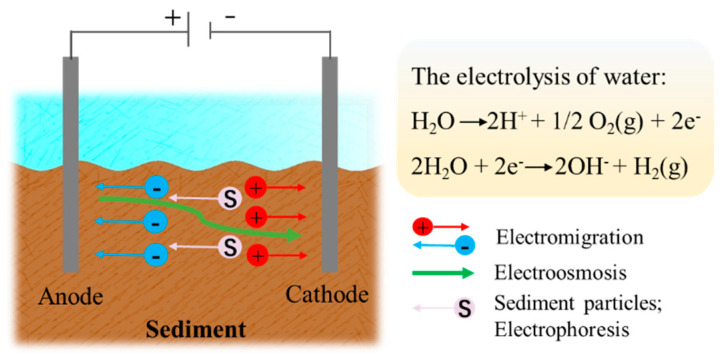
Schematic diagram of EKR (The most important electron transfer reactions at electrodes is the electrolysis of water).

**Table 1 ijerph-19-16767-t001:** Studies of in situ active capping and chemical amendments for heavy metal contaminated sediment in recent years.

Sediment	Adsorbent	Heavy Metal	Appling Method	Findings	Reference
The Hyeongsan River estuary, South Korea	Zeolite, AC/zeolite, AC/sand and zeolite/sand	Hg	Capping	Capping with AC/zeolite, AC/sand, and zeolite/sand reduced >90% of the Hg after 2 months.	[48]
Pudong New District, Shanghai, China	Apatite, apatite/calcite mixture	Cd	Capping	The reduction efficiencies of Cd by the apatite capping and apatite/calcite mixture capping on day 22 were 92.7% and 98.8%, respectively.	[49]
Lake Kivijärvi, Finland	BFS-GP granules	Fe, Zn, Ni, Cr	Mixing	The amendment effectively reduced the mobility of Fe, Zn, Ni, and Cr by about 50–90%.	[50]
The Gunneklev fjord, Norway	Lignite AC (A-AC, 5%) and activated BC (A-BC, 5%)	Hg	Mixing	The A-AC and A-BC amendments strongly reduced the available MeHg-concentration in porewater (by 87% for A-AC and by 93% for A-BC after 12 months).	[51]
A Baltic Sea bay, Sweden	Al, Polonite (calcium-silicate) and AC	Cd, Zn	Mixing	Al injection into anoxic sediments completely reduced the release of Cd (97%) and Zn (95%). Polonite mixed with AC reduced the release of Cd (67%) and Zn (89%).	[52]
A Former Mining Pit Lake, Arkansas, USA	Limestone, bentonite clay and gravel	Zn	Capping	A three-layer cap consisting of limestone (top) + bentonite clay (middle) + gravel (bottom) was the most effective.	[53]
Wulong River, China	BC and nano-Fe_2_O_3_ modified BC (nFe_2_O_3_@BC)	Cd	Capping	Both BC and nFe_2_O_3_@BC capping inhibited Cd release from sediment (reduction rates >99%), and nFe_2_O_3_@BC capping has better effectiveness.	[54]
An estuary pond within a former chlor-alkali plant, China	AC/bentonite, AC/kaolin and AC/montmorillonite	Hg	Capping	The caps with AC (3%) + bentonite (3%) and AC (3%) + kaolin (3%) reduced total Hg concentration in overlying water by 75–95% after 75-d operation.	[55]
The estuary of Sungai Kuala Perlis, Malaysia	Bentonite, kaolin and sand	Pb	Capping	Bentonite, kaolin, and mixture of bentonite with kaolin effectively reduced the release of Pb.	[56]
Guangdang River, Yantai, China	BC and BC-nanoscale zero-valent iron (nZVI/BC)	Cd	Mixing	BC and nZVI/BC reduced the released Cd concentrations by 31–69% and 26–73%, respectively.	[57]
Puhuitang Creek, Shanghai, China	Calcium nitrate and phosphate	Zn, Pb and Cu	Mixing	Over 50% of mobile Zn, Pb, and Cu might be reprecipitated in sediment.	[58]
The South River in Virginia, USA	Hardwood BC	Hg	Capping	80% of the Hg was retained on the biochar without promoting Hg methylation.	[59]
Xiangjiang River, China	Fe_3_O_4_, (α + γ)-Fe_2_O_3_, and αFe_2_O_3_	Cd	Mixing	(α + γ)-Fe_2_O_3_ exhibited better performances than the other iron oxides.	[60]
Maozhou River, China	CaCO_3_, Ca(OH)_2_, zeolite, kaolin, FeCl_2_	Cr, Ni, Cu	Mixing	Stabilization effect can be ordered as CaCO_3_ > zeolite > FeCl_2_ > kaolin > Ca(OH)_2_.	[61]
A mercury-contaminated site, USA	Mn(IV)-oxide phases pyrolusite or birnessite	Hg	Mixing	Reaction of Mn(IV) oxide with pore water should poise sediment oxidation potential at a level higher than favorable for Hg methylation.	[62]
A polluted reservoir, China.	Natural zeolite (N-zeolite)	Pb, Cd, Mn, Zn	Capping	The inhibition rates of Cd, Pb, Mn, and Zn were 35.7%, 85.7%, 65.6% and 57.8%, respectively.	[63]
Lake Pyhäjärvi and Lake Kivijärvi, Finland	BFS-GP, MK-GP, exfoliated vermiculite	Al, Cu, Fe, Cr, Zn, Ni	Capping	BFS-GP was suitable for Al, Cu, Fe and Ni; MK-GP for Cu, Cr and Fe; and vermiculite for Al and Zn.	[46]
The Yellow Sea, Korea	Dredged materials	Cr, Hg, Ni, Cu, Zn, Cd, Pb	Capping	The largest decreases were detected in Cr and Hg (≥ 80%), followed by Cd (74%), Cu and Zn (68%), Ni and Pb (10%).	[64]
Nanfei River, Hefei, China	Rice husk biochar (RHB)	Cu	Capping	RHB can maintain the concentrations of Cu below the national criterion at pH = 5 and 7.	[65]

**Table 2 ijerph-19-16767-t002:** Bioremediation for heavy metal contaminated sediment/soil in recent years.

Sediment/Soil	Biosorbent Type	Heavy Metals	Findings	Reference
Wangyu River, Jiangsu Province, China	*Hydrilla verticillata* and *Elodea canadensis*	Cd	The bio-concentration factors (BCFs) of both macrophytes exceeded 1.0. Two keystone bacteria (*Pedosphaeraceae* and genus *Parasegetibacter*) posed significant potential for promoting plant growth and tolerating Cd bio-toxicity.	[128]
Coastal sites along the Red Sea, Saudi Arabia	Mangrove plant	Cd, Cr, Cu, Ni, Pb, Zn	Sediment-to-plant transfer coefficient values were >1.	[129]
Wonorejo Estuary, Surabaya, Indonesia	Mangrove plant	Pb	The most effective mangrove involved in the accumulation of Pb was *Avicennia. alba* (BCFs: 1.13–90).	[130]
Lake Burullus, Egypt	Phragmites australis	Ni, Pb	The highest monthly Ni and Pb standing stock were 18.2 and 18.4 g/m^2^, respectively. The translocation factor of Ni and Pb was >1.	[131]
The coast of Rayong province, Thailand	Mangrove plant	Mn, Pb, Cr, Cu, Zn	The removal efficiency of heavy metals contaminated in sediment occurred in descending order of Mn > Pb > Cr > Cu > Zn (93.11%, 80.42%, 70.03%, 67.09% and 52.50%, respectively).	[132]
A wastewater pond, Philippines	Fugi (*Rhizopus* sp., *Mucor* sp. and *Trichoderma* sp.)	Cd, Cu, Fe, Zn	*Rhizopus* sp. was the most tolerant to all the heavy metals tested with the minimum inhibitory concentrations (MIC) of 5 mM < Cd ≤ 6.5 mM, 10 mM < Cu ≤ 15 mM, 30 mM < Fe ≤ 35 mM and 25 mM < Zn ≤ 30 mM.	[133]
Yuepu industrial area, Shanghai, China	Fugi (*Fusarium fujikuroi*, *Fusarium solani*, *Trichoderma citronoviridae* and *Trichoderma reese*)	Cd, Cr, Cu, Pb, Hg, Ni	The highest biosorption capacity of Pb was exhibited by *Trichoderma citronoviridae*, while *Trichoderma reesei* showed the best absorption capacity of Cu, followed by *Fusarium solani*.	[134]
The Lerma-Chapala Basin, Mexico	Bacteria (mainly including *Delftia* and *Pseudomonas*)	Zn, As, Ni	The bacteria showed high heavy metal resistance, especially to Zn, As and Ni, which could be employed in the bioremediation process.	[135]
Sangan iron ore mine, Iran	Cyanobacteria (*Oscillatoria* sp. and *Leptolyngbya* sp.)	Cr, Fe, Ni, As, Pb, Cu	Cyanobacteria inoculation decreased the available concentration of Pb and Ni. The maximum metal removal efficiency was 32%.	[136]
Kitchener Drain, Nile Delta	*Eichhornia crassipes*, *Ludwigia stolonifera*, *Echinochloa stagnina*, *Phragmites australis*	Cd, Pb, Ni	*Phragmites australis* accumulated the highest concentrations of Cd (57.5 mg/kg) and (109.0 mg/kg), while *Eichhornia crassipes* accumulated the highest concentration of Pb (277.4 mg/kg).	[137]
Shipbreaking area, Bangladesh	Mangrove plant	Zn, Pb, Cu, Cr	*Acanthus ilicifolius* showed hypermetabolizing capabilities for most metals, and *Avicennia alba* showed hypermetabolizing capabilities for Cu, Zn, and Fe.	[138]
Jaran Bay and Onsan Bay, Korea	*Seagrass Zostera marina*	Cd, Zn, Hg	*Zostera marina* transplants accumulated a great amount of heavy metals in their tissues, which have the phytoremediation potential for the heavy metal-contaminated sediments.	[139]
The western watershed, Thailand	EDTA and diethylenetriamine pentaacetic acid (DTPA) combined with *Water Hyacinth*	Cd	*Water hyacinth* accumulated Cd of 112.73 mg/kg in root within 3 months.	[140]
Suyeong Bay, Korea	*Phaeodactylum tricornutum*, *Nitzschia* sp., *Skeletonema* sp., and *Chlorella vulgaris*	Cu, Zn	*Chlorella vulgaris* grew under red LED and exhibited the highest Cu and Zn removal capacities with values of 17.5 × 10^−15^ g Cu/cell and 38.3 × 10^−15^ g Zn/cell, respectively.	[141]

## Data Availability

Not applicable.

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
