# Peer review of "In Situ Remediation Technology for Heavy Metal Contaminated Sediment: A Review"

_ijerph, 2022, doi:10.3390/ijerph192416767_

Round 1

Reviewer 1 Report

General comments

The manuscript “In-situ Remediation technology for heavy metal contaminated sediment: A review” summarized the state-of-art in-situ remediation technology for contaminated sediment. The content covered the development of water diversion, capping, electrokinetic remediation, chemical amendments, bioremediation and combined remediation, but the description of the latest technology was not comprehensive. The English language and format of this paper also needs to be polished further.

Specific comments

1.     Line 106-107, the sentence ‘On the other hand…affects metals fate’ has grammatical mistake.

2.     Section 3.1 electrokinetic remediation, frontier technology progress should be mentioned.

3.     Line 154-160, the description of water diversion is simple. For example, factors influence the improvement of water quality following water diversion and models can be added.

4.     Section 3.2, widely used amendments also include silicocalcium materials, iron-bearing materials and aluminum salts, and so on.

5.     Give more information about the toxicity or risk of chemical amendments.

6.     Line 242, the characteristics of hyperaccumulators (such as the criteria used for hyperaccumulation varies by metals…) should be more detailed

7.     Advantages and disadvantages of phytoremediation, for example, phytoremediation costs are expected to be < 25% of some other remediation techniques (Mani and Kumar, 2014); limitations: low biomass, slow growth rate, long timeframes and so on.

Mani D, Kumar C (2014) Biotechnological advances in bioremediation of heavy metals contaminated ecosystems: an overview with special reference to phytoremediation. Int J Environ Sci Technol 11(3):843–872

8.     Line 304-306,Physical remediation Chemical-biological combined remediation contains phyto-stabilizing agent combined remediation and phyto-activator combined remediation, which promotes the processes of phytostabilization and phytoaccumulation.’ This sentence is not clear, delete Physical remediation?

9.     There are some formatting errors in reference, e.g. the journal name.

Author Response

Response to Reviewer 1 Comments

Comments and Suggestions for Authors

General comments

The manuscript “In-situ Remediation technology for heavy metal contaminated sediment: A review” summarized the state-of-art in-situ remediation technology for contaminated sediment. The content covered the development of water diversion, capping, electrokinetic remediation, chemical amendments, bioremediation and combined remediation, but the description of the latest technology was not comprehensive. The English language and format of this paper also needs to be polished further.

Specific comments

  1. Line 106-107, the sentence ‘On the other hand…affects metals fate’ has grammatical mistake.

This sentence has been corrected in revised manuscript.

 On the other hand, organic matter provides a food source for microorganisms and indirectly affects metals fate. (Line 106-107)

  1. Section 3.1 electrokinetic remediation, frontier technology progress should be mentioned.

   On the other hand, EKR combined other remedy techniques are more effective, for example, EKR-bioremediation requires low energy and improves the growth of plants and the spread of microorganisms; EKR-PRB simultaneously achieves pollutants removal, degradation or recycling from soil/sediment [69, 77]. In terms of improving energy utilization efficiency and developing self-powered technology, pulsed electric fields (FE), solar-power and microbial fuel cells have been extensively studied [69]. (Line 160-166 in revised manuscript)

  1. Line 154-160, the description of water diversion is simple. For example, factors influence the improvement of water quality following water diversion and models can be added.

   Water diversion is to introduce clean water to polluted areas, so that contaminants concentrations are diluted and water self-purification process is accelerated [78]. There are many factors affecting this process, such as diverted discharge, diversion routes, wind direction/magnitude and ways of managing diversion projects [79, 80]. Generally, long-term diversion with a low flow rate is better than short-term diversion with a high flow rate, and a low nutrient concentration of diverted water is also important [79-81]. Hydrodynamic and water quality models are introduced to simulate the transport of pollutants, the flow movement , water level changes and the effectiveness of water transfer projects, including HEC-RAS, MKIE11, MIKE 21 FM, EFDC, FVCOM and other models [80, 82-86]. This technique improves water exchange and the reoxygenation rate, especially suitable for polluted lakes [82]. Successful examples are Moses Lake, Xihu Lake, Tianyinhu Lake and Dongshan Lake [79, 82, 87, 88]. Water diversion responds quickly to pollutants, but the diverted water needs to be reasonably distributed to avoid the occurrence of dead water zones [79]. (Line 170-185 in revised manuscript)

  1. Section 3.2, widely used amendments also include silicocalcium materials, iron-bearing materials and aluminum salts, and so on.

   Furthermore, amendments of silicocalcium materials (e.g. CaO and MgO, etc), iron-based materials (e.g. Fe0, Fe2O3, Fe3O4 and Fe(OH)3, etc), aluminum salts (e.g. aluminum chloride and aluminum polychloride, etc), industrial residue (e.g. steel slag, etc) and polymers (e.g. alkali-activated materials and biopolymers, etc) also can reduce the bioavailability of metals effectively, applying as pilot- or full-scale in sediment remediation [12, 43, 103, 104]. (Line 219-225 in revised manuscript)

  1. Give more information about the toxicity or risk of chemical amendments.

 Excessive or improper use of phosphate may lead to water eutrophication and other risks [91]. (Line 201-202 in revised manuscript)

At the same time, modification adds additional costs and new chemical agents also increase environmental threats, for instance, organoclays between 5 and 100% v/v have adverse effects on crustaceans [93]. (Line 207-209 in revised manuscript)

As a new type of environmental remediation materials, nanomaterials have great uncertainties and should be used with caution. Nanomaterials can easily enter the environment and living cells due to their tiny size. And that the nanomaterials used to remedy sediments are not easily separated after restoration, resulting in secondary pollution and adverse effects on benthic microbial communities and aquatic organisms [109]. (Line 244-249 in revised manuscript)

  1. Line 242, the characteristics of hyperaccumulators (such as the criteria used for hyperaccumulation varies by metals…) should be more detailed

 The criteria used for hyperaccumulators are > 100 mg/kg for Cd; > 1000 mg/kg for Cu, Ni, and Pb; > 10,000 mg/kg for Mn and Zn in plant shoots (all accumulations are dry weight) [12]. (Line 286-288 in revised manuscript)

  1. Advantages and disadvantages of phytoremediation, for example, phytoremediation costs are expected to be < 25% of some other remediation techniques (Mani and Kumar, 2014); limitations: low biomass, slow growth rate, long timeframes and so on.

Mani D, Kumar C (2014) Biotechnological advances in bioremediation of heavy metals contaminated ecosystems: an overview with special reference to phytoremediation. Int J Environ Sci Technol 11(3):843–872

 It is an operationally simple, cost-effective (25% less than other remediation techniques) and a promising clean-up solution for a wide variety of contaminated sites despite some restrictions (e.g. the climatic and geological conditions, low biomass, slow growth rate, etc) [111]. (Line 256-259 in revised manuscript)

  1. Line 304-306, ‘Physical remediation Chemical-biological combined remediation contains phyto-stabilizing agent combined remediation and phyto-activator combined remediation, which promotes the processes of phytostabilization and phytoaccumulation.’ This sentence is not clear, delete Physical remediation?

 “Physical remediation” has been deleted. (Line 354 in revised manuscript)

  1. There are some formatting errors in reference, e.g. the journal name.

Reference formatting errors have been corrected in revised manuscript.

Reviewer 2 Report

Manuscript ID: ijerph-2070371

Title: In-situ Remediation technology for heavy metal contaminated sediment: A review

The manuscript describes the leading technologies employed in heavy metal elimination, emphasizing those employed in sediment treatment. Physical, chemical, biological, and combined remediation technologies were reviewed, highlighting their advantages and disadvantages. The manuscript overall is well organized, and relevant information is presented. Only minor errors were observed in the manuscript. However, it could be of high value to include figures illustrating the main technologies applied in sediment heavy metal remediation, as well as at least a table summarizing recent studies applying reviewed technologies in the treatment of sediments, including information on the type of treatment, efficiency, or amount of heavy metals removed. The manuscript could be improved, by including figures and tables to support the importance of sediment heavy metal remediation technologies reviewed.

Additional commentaries are below:

In the whole manuscript use italics for “in-situ” and “ex-situ”

Line 22, change “7% - 8%” by “7-8%”

Line 24, in the fragment “…steel was the world's largest producer in 2009…”, please change the word “producer” to give sense.

Lines 31-33, please review the redaction, it is not clear

Lines 41-42, add “arsenic exposure can result…”

Line 45, “endocrine” could be “endocrine system”

Line 47, in the fragment “…Heavy metals cannot be effectively biodegraded through environmental media…”, please review due to heavy metals cannot be biodegraded in any way.

Line 50, please review the form of the “to be” verb, maybe change “are” by “have been”

Line 87, add a space in “carbonates[23]”

Line 138, add a space in “reported[44]”

Line 220, add a space in “medias(e.g. sediment)”

Line 222, add a space in “predict[12]”

Line 224, eliminate extra space in “lization,phytostabilization”

Line 254, define the acronyms “EDDS and EDTA”

Round 2

Reviewer 2 Report

The authors addressed all the reviewer’s suggestions in the manuscript, I consider the manuscript suitable for publication